# Analysis of Particle Variation Effect on Flexural Properties of Hollow Glass Microsphere Filled Epoxy Matrix Syntactic Foam Composites

**DOI:** 10.3390/polym14224848

**Published:** 2022-11-10

**Authors:** Olusegun Adigun Afolabi, Krishnan Kanny, Turup Pandurangan Mohan

**Affiliations:** Mechanical Engineering Department, Durban University of Technology, Durban 4000, South Africa

**Keywords:** syntactic foam, particle variation, flexural properties, volume fraction, scanning electron microscope

## Abstract

Syntactic foam made from hollow glass microspheres (HGM) in an epoxy matrix has proven to be a good material with a strong structural strength. Understanding filler particle size variation is important in composite material formation, especially in syntactic foam, because of its numerous applications such as aerospace, marine, and structural purposes. In this present work, the effects of particle variation in different sizes (20–24 µm, 25–44 µm, 45–49 µm, and 50–60 µm) on the mechanical properties of the syntactic foam composites with a focus on flexural strength, modulus, and fracture surfaces are investigated. The particle sizes are varied into five volume fractions (5, 10, 15, 20, and 25 vol%). The results show that the highest flexural strength is 89 MPa at a 5 vol% fraction of 50–60 µm particle size variation with a 69% increase over the neat epoxy. This implies that the incorporation of HGM filler volume fraction and size variation has a strong effect on the flexural strength and bending modulus of syntactic foam. The highest particle size distribution is 31.02 at 25–44 µm. The storage modulus E’ increased at 30 °C, 50 °C, and 60 °C by 3.2%, 47%, and 96%, respectively. The effects of wall thickness and aspect ratio on the size of the microstructure, the fracture surfaces, and the viscoelastic properties are determined and reported accordingly.

## 1. Introduction

Syntactic foam composites (SFC) made of hollow glass microsphere/epoxy resin (HGM/EP) are an excellent material in aerospace and structural applications due to their lightweight nature [1,2,3,4]. Syntactic foams exhibit a special characteristic combination of physical and mechanical properties suitable for marine applications, such as high specific stiffness, high strength, low density, energy absorption capacity, good impact resistance, closed-cell microstructure, and low water absorption capacity [2,3,5,6]. Modern technology requires a better composite material than metals, wood, ceramics, glass, aluminum, etc. due to their complexity and rigidity. The need for composite materials arose to improve upon the existing materials to achieve high and better quality performance. Syntactic foam’s low-density capacity has enlisted it as a good drilling fluid in oil and gas or deep-sea exploration because of its floatability on water. They are employed as cores in sandwich materials, which are applicable in marine structural concepts, jet engine parts, and ship bodies because of their thermal stability, high bending stiffness, low moisture absorption, and lightweight nature [3,7,8,9,10].

Different matrices have been employed in the fabrication of syntactic foam, which can be natural or synthetic in nature, ranging from organic to inorganic particulate polymer materials [11,12,13,14]. The good curing property possessed by epoxy resin has made it a choice among polymer matrices in composite materials, as reported by many researchers [15,16,17,18,19]. Arpitha et al. [20] used epoxy (LY-556) as the matrix in studying the mechanical properties of hybrid sisal-silicon carbide (SIC)-glass composites. The result shows that the epoxy-based sisal and glass composites with SIC at 3% gave better flexural properties than the composites without SIC because of the variance in their physical properties. Gupta et al., and Huang et al., confirmed that HGM/EP possesses a good interfacial bond, which improves the mechanical properties of syntactic foam [19,21]. The flexural behavior of syntactic foam in terms of bending stiffness and strength has been investigated based on sandwich beams [7], fiber-reinforced composites [22], hollow glass bead-filled composites [23], and glass microballoons [24,25]. Results of these findings show that syntactic foams’ flexural and specific modulus are higher than the neat epoxy, while the flexural strength reduces with an increase in volume fraction of fillers such as glass beads, cenospheres, microspheres, etc. [24,25,26,27]. Huang et al. [28], reported an opposite variation tendency in the properties of syntactic foam at room temperature and high temperature. It was recorded that the flexural strength at room temperature decreased with increasing HGM volume fraction, while at a temperature of 400 °C, 600 °C, and 800 °C, the flexural strength increased with increasing HGM content.

The dynamic mechanical analyzer has been widely used as a technique for viscoelastic material characterization parameters [29,30,31]. This technique entails applying a sinusoidal force on the specimen to obtain a phase difference in stress and strain values as a response to viscoelasticity. The values are recorded as storage modulus (E′) and loss modulus (E″). The storage modulus is the amount of energy stored in the material, while the loss modulus refers to the energy dissipated by the material in each cycle of the sinusoidal deformation. The damping parameter, Tan δ is the ratio of the loss modulus to the storage modulus [21,30,32].

Most of the previous literature used multiple types of HGM in their fabrication of syntactic foam. Swetha et al. [33] used three types of HGM, K15, S22, and K46, in fabricating their syntactic foam. They reported a decrease in strength and modulus with an increasing volume fraction of the microsphere as the wall thickness of the microsphere increased. Colloca et al. [34] also used the following two types of HGM: S220 and K460. The report shows that the specific properties of syntactic foams are considerably higher than those of carbon nanofiber/epoxy composites. Ullas et al. [35] also utilized K15 and K46 as their HGM fillers. They concluded that loading microballoons with a high-volume fraction ≥ 60% resulted in too high viscosities that prevented solvent processing.

Based on the available literature and facts established above, there is a need to understand the input of the particle sizes variation of HGM on the mechanical and structural properties of syntactic foam as few works have been reported on it [36,37]. Moreover, the strength of the syntactic foam is largely dependent on the strength and size of HGM [38]. Chen et al. [36] studied the effects of three different size distributions of HGM (22.5~56 µm; 56~88 µm and 92~125 µm) on the compressive strength of syntactic foam. Moreover, Al-Shalani et al. [38], recently studied the effect of expanded glass particle size on the metal syntactic foam (MSF). They reported that the highest crushing strength resulted in an elastic stiffness and yield stress increase. Moreover, Kishore et al. investigated the effects of two size ranges (65~100 µm and 44~175 µm) on the compressive strength and modulus of syntactic foam [37]. There is a relative absence of systematic studies on smaller particle sizes and a single type of HGM due to the larger size distribution and higher volume fractions studied in previous works where the particle sizes tend not to affect the mechanical properties of the syntactic foam [36,37,38,39]. This research gap must be investigated and served as a novelty in this present work because most previous studies compared several HGM types to the mechanical properties of a syntactic foam. Therefore, there is room to explore the use of single HGM type with an improved processing method and better mechanical performance on the syntactic foam. This study focused on the effects of different particle sizes, aspect ratios, and wall thickness on the flexural strength and modulus properties of syntactic foam composite. The viscoelastic properties and fracture mechanisms of the failure mode of the SFC at each size variation were also discussed. The SFC can be used as the core in sandwiches and for structural purposes. Therefore, their lower fracture strength and strains do not limit their applicability in marine vessels and aerospace materials.

## 2. Materials and Methods

### 2.1. Materials

Epoxy resin LR 20 and hardener LR 281 purchased from AMT composites were used as matrix and binder. T60 microballoons filler was procured from Anhui Elite Industrial Co., Ltd., Hong Kong Elite Industrial Group Ltd., 18F, Block C, Winning Ocean Plaza, No. 680 South Maanshan Road, Hefei 250051, China. Single type of glass hollow microsphere (HGM) was used to study the effect of high-density microsphere as it affects the changes in volume fraction. Resin and hardener were mixed in a ratio 10:3 by weight. The detailed characterization of HGM has been reported earlier by the author [40].

### 2.2. Fabrication of Syntactic Foam Composites

SFC samples for the flexural experiment were prepared under room atmospheric conditions (30 °C). The detailed method of fabrication has been published in the previous report by the author [40]. The solution was then poured into a silicon mold for casting and left to cure for 16 h. The syntactic foam slab was then post-cured for 4 h at 70 °C. This process was performed for all five (5–25 vol%) volume fractions and the whole process was repeated for all four size variations. Figure 1 shows the schematic representation of the size distribution in processing the SFC, which Figure 2 shows the processing sequence employed in fabricating the SFC. Detailed fabrication procedure and process of separation into five volume fractions has been reported earlier by the author [40].

### 2.3. Particle Size Distribution Analysis

The size distributions of the four particle ranges for HGM were characterized by particle size analyzer using PSA 1190 from Aaton Paar, Durban, the values are listed in Table 1. The D10 indicates that 10% by volume of the samples falls within the range of the values under it. Likewise, D50 and D90 indicate that 50% and 90% by volume of the samples fall within the ranges of the values under it. The volume and geometric properties of all the four-particle size variations are listed in Table 2. The wall thickness and aspect ratio define hollow geometry of the HGM. The particle size enclosure also plays an important role in the severity of the HGM reaction [41].

### 2.4. Design Experimentation and Specimen Coding

The HGM was varied into four different particle sizes using sieves (i.e., 20–24 µm “AA”; 25–44 µm “BB”; 45–49 µm “CC” and 50–60 µm “DD”) to investigate their effect on the mechanical properties of syntactic foam. Further, particle size analysis (PSA) using Anton Paar PSA 1190 particle size analyzer was conducted on the varied HGM sizes. This was to determine the wall thickness (ω), particle size distribution (V_d_), average particle diameter (d) of strength distribution of HGM, and aspect ratio (a) as a function of the varied particle sizes. In order to investigate the relationship between the aspect ratio “a” and the strength of HGM [39], the aspect ratio “a” was given as (d/ω) and values of “ω”, “V_d_”, and “d” were given in Table 2. Single-type HGM filler was used for the manufacturing of the SFC at each size variation and varied into five volume percentages with varying volumes of matrix. It can be seen from Table 2 that BB has the largest V_d_, which shows that the size ranges is more uniform and influenced their mechanical properties in volume distribution. The ratio of matrix to filler in each specimen is coded as XXT60-OOYY. XX represents EP (epoxy-based) SFC, T60 is the trade name of the HGM filler, OO represents the particle sizes and YY represents the volume fraction of the HGM in each composition of SFC. For example, EPT60-AA5 represents the SFC with 5 vol% of T60-HGM filler for 20–24 µm particle size variation.

### 2.5. Flexural Test

MTS 793 servo-hydraulic machine with a load cell of 100 KN and test speed of 2 mm/min was used. The experimental set-up was performed in a way that one-point loading head was hung up in between the jaws of the testing machine and a two-point support fixture was beneath. A constant span length of 48 mm (which represents the standard ratio of span length to thickness e.g., 16:1) was maintained during the test for all the samples of dimensions 125 *×* 14 *×* 3 mm^3^ (i.e., length, width, and thickness, respectively). Three (3) samples were tested per specimen according to ASTM D790-01 as shown in Figure 2. All the specimens experienced fractures as the crosshead was maintained at strain of 5 mm/mm and the results data were collected after the test for analysis.

### 2.6. Scanning Electron Microscopy (SEM), Thermogravimetric Analysis (TGA), and Dynamic Mechanical Analysis (DMA)

Microstructural investigations were conducted on the broken samples to observe the fracture mechanisms and surface morphology of the filler as regards the effect of size variations on the fracture surfaces. This test was achieved with the aid of scanning electron microscopy (SEM). The fractured flexural specimens were examined using a Zeiss EVO 1 HD 15 Oxford instrument X-max scanning electron microscope (SEM). The specimens were gold coated before the SEM was conducted because the syntactic foams are not conductive and there is need for the flow of electrons for the fracture images to be seen properly, this was performed by Quorum Q 150R ES machine for 6 min. The TGA and DMA were conducted using TA Model Q800 V20.6 Dynamic Mechanical Analyzer instrument.

## 3. Results and Discussion

### 3.1. Flexural Properties of Syntactic Foam Composites

Figure 3a–d shows the stress-strain graphs of the SFC with varied HGM particle sizes compared with the neat epoxy. All the samples were brittle in nature and broke after reaching maximum yield stress. They also possess better bending capacity than the EPT60-0 except for EPT60-AA10 and EPT60-CC25 with slightly lower stress values. This rare phenomenon is a possibility of a poor interaction between the filler and the matrix during mixing, leading to early breakage and reduced strain of the specimens. Moreover, it might have resulted from interface debonding and matrix cracking of the specimens leading to early failure and reduced strain at break in Figure 3a, the highest stress value was obtained at EPT60-AA20, while in Figure 3b–d, the highest stress was at EPT60-BB5, EPT60-CC5, and EPT60-DD5 respectively. An increase in flexural stress at 5 vol% of HGM as the size variation increases can be related to the fact that the addition of a smaller quantity of HGM with a high aspect ratio “a” gives better dispersion in the matrix during the mixture, resulting in high strength and stiffness [29].

Table 3 shows the flexural modulus of SFCs for the varied HGM fillers. The neat epoxy, being the base material for comparison, has a higher flexural modulus than all the homogeneous SFCs. However, at AA, EPT60-20 has the highest flexural modulus of 1.02 GPa. At BB, the highest flexural modulus is 1.09 GPa at EPT60-5. At CC and DD, the highest flexural modulus is 1.31 GPa and 1.14 GPa at EPT60-5, respectively. The increase in flexural modulus of the SFCs at higher particle sizes was a result of stiff particles in HGM, reduced wall thickness “ω”, and high aspect ratio “a” at lower volume fractions. It could also be a result of density reduction as the HGM volume fraction increases [25]. Moreover, it shows that the particle sizes of HGM affected the mechanical properties of syntactic foam, which corresponds to the report by Al-Sahlani et al. [38] where the varying sizes of the particles change the microstructure and flexural modulus of glass-metal syntactic foam.

Table 4 shows the flexural strength (MPa) of SFC for the homogeneous HGM filler. The flexural strength of the SFCs increased compared with the NE except at EPT60-25CC with a 2.6% decrease. At AA, the SFC increased in flexural strength for all the volume fractions of HGM. The highest flexural strength is 80.7 MPa at EPT60-20AA, which is a 50.8% increase compared to the neat epoxy. At BB, the SFCs show a trend of a decrease in flexural strength with an increasing volume fraction of HGM. The highest V_d_ of 31.02% at BB might have also influenced this trend. This possibly implies that the uniformity of particle distribution reduces the flexural strength with increasing volume fraction. The highest flexural strength was 86.7 MPa at EPT60-5BB, which is a 62.1% increase compared to the NE. The reduced flexural strength at an increased volume fraction of HGM is attributed to the fact that a higher concentration of HGM filler in the matrix reduces the adhesion interface between the filler and the matrix. Thereby resulting in low strength and brittle structure [25]. At CC, the highest SFC is 85.5 MPa at EPT60-5CC, which is a 59.8% increase compared with the NE. The increased strength at lower volume fractions and higher particle sizes can be attributed to higher “a” and smaller wall thickness “ω”.

At DD, the highest SFC flexural strength is 89 MPa at EPT60-5DD, which is a 66.4% increase compared with the NE. It doubles as the highest flexural strength of all the SFCs, both at volume fraction variations and particle size variations. As the particle size increases, the highest flexural strength also increases for BB, CC, and DD, which is at a 5vol% fraction of HGM. This was because the addition of a smaller quantity of filler particles has been observed to give a better dispersion in the matrix, lower agglomeration, and reduced porosity content, resulting in high strength and stiffness. A similar observation was reported by Bharath et al. [6] where the strength of syntactic foam increased with an increasing volume fraction of HGM compared to the neat matrix. It is also an indication that the flexural strength is reliant on the microballoons volume fraction and particle sizes [3]. Garcia et al. [27] also reported an increase in modulus of 11–14% for both syntactic foams at room and artic temperature conditions. It was further observed that reduced “ω” resulted in a higher “a”, which invariably increased the modulus and strength of SFCs. This trend was noticed majorly at lower volume fractions with respect to the higher particle size variation of HGM.

Table 5 illustrates the influence of the HGM size variations and volume fractions on the flexural strain at the break of the SFC. Flexural strain, or elongation at break, is obtained during tensile testing. Flexural strain at break is an important parameter used in characterizing the flexural fracture toughness of SFCs. The chain mobility was responsible for the flexural break of SFC during flexure testing and is controlled by the concentration of the filler and their forming network within the polymer formation of the matrix, making them lower than that of the pure NE [42]. The decrease in the tensile strain of SFC at break can be related to the presence of large aggregates and a strong cluster of HGM filler at higher volume fractions in the SFC. It can also be said that the tensile fracture of SFC can be modified by the content of the HGM filler at a lower concentration. This can further influence the formation of stress concentration in the interfacial layer between the matrix and the HGM filler [23].

### 3.2. SEM Micrographs of Syntactic Foam Composite

Figure 4a, Figure 5a, Figure 6a and Figure 7a [43] present the plain fracture migrograph of epoxy resin, where plain rough surface and matrix porosity are evident. While Figure 4b–f, Figure 5b–f, Figure 6b–f and Figure 7b–f represent the flexural fractured micrograph of SFC with homogeneous HGM filler. From Figure 4c,d, the micrograph of the fractured HGM was brittle, which occurred due to much agglomeration between the particles of the microspheres and the matrix. The agglomeration occurred due to air entrapped during the mixture of HGM and the epoxy resin, which is evident on the surface of the morphology by the smaller microsphere, but it does not affect the homogenous distribution of HGM in the matrix. It could also be due to their smaller sizes causing a smooth separation during fracture and, in turn, increased flexural strength.

From Figure 4d–f, ductile fracture occurs as a result of good adhesion and good interface interaction between the HGM and epoxy resin [25]. The fractured micrographs were seen on the rough morphology surface with some microsphere debris, which could be attributed to the ductile fracture under flexural loading conditions and resulted in high flexural strength. From Figure 5b, Figure 6b and Figure 7b, a rough morphology surface can be seen with little agglomeration, which can be possibly due to small vol% (5 vol%) of HGM in the SFCs though with larger particle sizes (25–44 µm), (45–49 µm), and (50–60 µm), respectively.

It can also be said that good surface interaction occurred between the micrograph of the HGM and the epoxy resin with little agglomeration, resulting in higher flexural strength. On the other hand, Figure 5c–f, Figure 6c–f and Figure 7c–f present surface micrographs of SFCs, which revealed fractured microspheres, deboned microspheres, matrix porosity, and agglomeration. There was an increase in the debris on their surfaces, which was possible due to an increase in the HGM vol% in the SFCs. They exhibited ductile fracture, which is consistent with the fact that the adhesive force between the HGM and the epoxy matrix weakens during flexural loading, resulting in their reduced flexural strength.

It can generally be said that a good interface interaction occurs between the HGM and the matrix for all the size variations. At the bigger sizes and small volume fractions (5 vol%) of HGM, the SFCs were able to withstand flexural loading, possibly due to smaller ω and higher “a” than the smaller sizes and larger volume fractions (10–25 vol%), resulting in their improved flexural modulus, strength, and strain values. Further observation revealed that the ductile fracture experienced by the larger sizes is relative to the microsphere’s nature. However, the flexural strength, strain, and modulus decreased with an increasing volume fraction of HGM due to stronger interfacial strength experienced during flexural loading. This was because flexural properties are known to strongly require the interfacial bonding characteristics to transfer the load from the matrix to the HGM particles [5,22,25,44,45,46]. Moreover, it can be said that the micrograph clearly illustrates that the fabrication process has resulted in the impeccable interface relationship of the specimen [47].

### 3.3. Thermal Analysis

#### 3.3.1. Thermogravimetric Analysis (TGA)

Figure 8 shows the thermal temperature compositions of SFC at homogeneous HGM filler sizes compared to neat epoxy. At 5% weight loss, the degradation temperature of all the SFCs at AA, BB, CC, and DD was reduced by 12% compared to the neat epoxy. This occurred due to possible weakness in the cohesive force of homogeneous HGM size dispersion at 5% degradation. It can also possibly be due to poor cross-linking of the functional group of the filler molecules during covalent bonding with the matrix, resulting in low molecular weight of the SFCs at 5% weight degradation. Salleh [48] reported an increase in the degradation of pure vinyl ester than the syntactic foam composites due to much more debris or flakes from the glass microballoons. Meanwhile, at 20% and 55% weight loss, the SFCs at AA, BB, CC, and DD shows higher thermal stability of 4.5% and 2.7% respectively than the NE. This is possible because the chemistry of the epoxy resin influenced a greater interaction with the inorganic filler, thereby promoting the increase in the thermal stability of the SFCs [49] because high-temperature treatment can damage the properties of a material [50].

#### 3.3.2. Dynamic Mechanical Analysis

##### Storage Modulus

The storage modulus (E′) values for the SFCs and NE versus temperature containing various volume fractions and size variations are shown in Figure 9a–d. The E′ of the SFCs measures the stored energy in the elastic portion of the composite. The E′ at all forms of SFCs decreases in value with temperature increase and shows a drastic drop around the glass transition region. When compared to the NE, the E′ is higher only at EPT60-AA25 and EPT60-CC25. Other SFCs decrease the value of the E′ than the neat epoxy. A similar observation was reported by Huang et al. [21], where a decrease in storage modulus occurred for all types of syntactic foams compared to the neat epoxy. The decline of E′ is due to the incorperation of HGM, which makes the matrix more brittle. At a temperature above 80 °C, the curves reach the flow region where E′ stabilizes to a very low value and the variation is negligible.

At temperatures around 50 °C and 60 °C, the E′ for all the SFCs samples improved compared to the NE, with the highest E′ values at EPT60-AA25, EPT60-BB5, EPT60-CC25, and EPT60-DD10 and increased by 1153 MPa, 860 Mpa, 669 Mpa, 1040 Mpa, and 875 Mpa, respectively. The improved low-temperature E′ values of the SFCs are attributed to the restricted movement of the polymer chain as a result of good interaction between HGM and NE during mixing. Poveda, et al. [31] reported similar significantly higher E′ of 7.9%, 14.6%, and 400% at temperatures of −50 °C, 30 °C, and 175 °C, respectively, compared to the NE. The improved E′ is an indication that the HGM incorporation into the neat epoxy can reduce the viscosity flexibility of SFCs, leading to increased mechanical properties of SFCs.

The improved E′ values compared to NE at 50 °C and 60 °C can be attributed to good interaction and interfacial bonding between the HGM and NE molecules during mixing, resulting in increased flexural strength of SFC mostly at lower volume fraction and higher particle sizes [29,51].

##### Loss Modulus

Figure 10a–d shows the loss modulus (E″) of the SFCs, which is the viscous response and a measure of the energy dissipated at heat per cycle under the deformation of the composites. The E″ value for SFC with heterogeneous HGM decreases compared with the EPT60-0, except at EPT60-25, which increased by 2 MPa than the EPT60-0. This shows that at room temperature, SFCs dissipate more energy into the atmosphere, which affects their loss modulus.

At 30 °C and 50 °C, the E″ decreases compared to the NE for all the SFCs because of good energy dissipation in the process of forming the composites. The occurrence of reduced E″ at 30 °C and 50 °C could be responsible for the reduced flexural modulus of SFCs compared to the NE. Moreover, at reduced temperatures, the composite’s constituents tend to increase molecular mobility and thus lose their tight packing structure, which progressively leads to a decrease in E″ at the rubbery region [29,52]. A similar observation was reported by Shunmugasamy et al. [30] where the E″ of the neat resin is higher when compared to various types of the SFC. The increase in “ω” increased the E″ values of SFCs except at DD, which does not align with this trend. This shows that the E″ of syntactic foam was noticeably affected by the particle sizes of HGM. At 60 °C, the E″ values increased compared to the NE for all the SFCs samples. The improved E″ values compared to NE can be attributed to good interaction and interfacial bonding between the microsphere and the matrix during mixing, resulting in increased flexural strength.

##### Tan Delta

Figure 11a–d shows the damping factor (Tan δ), which is the ratio of E″ and E′ and is a measure of impact and elastic characteristics or the damping capability of the SFCs, the peak height is closer to the energy dissipation of the SFC. The glass transition temperature (T_g_) of the SFCs can also be evaluated from the peak temperature of Tan δ. The variation of tan delta (δ) of SFCs for the homogeneous HGM is estimated from Figure 11a–d. The T_g_ is determined from the tan δ peak at a temperature of above 70 °C, which increased compared to the NE up to 86 °C. The addition of HGM into the epoxy matrix caused the peak of Tan δ to shift towards an increased temperature, which could be attributed to the brittle nature of the SFC, which orchestrated the restrictions in the movement of epoxy molecules [52]. The height of the tan delta peak is an indication of the degree of molecular mobility, which is raised when the concentration of HGM volume fraction and “ω” increases and the shape of the peaks becomes wider [21,32]. This is possibly due to the reduced intensity in the intermolecular chain during the mixture between the HGM and the epoxy matrix [29].

## 4. Conclusions

The effect of particle size variation on the mechanical properties of HGM/EP syntactic foam composites was investigated in this study. From this study, the following conclusions were drawn:

The particle size distribution (V_d_) was highest at BB with 31.02%. This shows that HGM distribution at particle size ranges is more uniform than the others;The results showed that flexural stress and strain of the syntactic foam composites increased with the inclusion of the HGM filler more than the neat epoxy for all the size variations and volume fractions except at EPT60-AA10 and EPT60-CC25, which can be considered insignificant because of the small percentage difference, i.e., 1.6% and 2.6% respectively less than the neat epoxy;The flexural modulus of all SFCs decreased by 45% compared to the neat epoxy. However, the flexural modulus increased with increasing particle size variation, which can be related to increasing wall thickness “ω” with increasing particle sizes;The highest flexural strength is 89 MPa at EPT60-DD5, which is a 66.4% increase compared to the neat resin at 5 vol% fractions and 50–90 µm particle size variation. This is an indication that the flexural strength increased with an increase in size variation and volume fraction in the syntactic foam composites;The microstructural analysis on the fracture surfaces revealed the size of the filler HGM at each variation through a higher magnification (1500×) scanning, while agglomeration, broken and unbroken HGM on the fractured surfaces at each particle size variation could be clearly seen at a lower magnification of micrographs (200×);Moreover, reduced wall thickness “ω” resulted in a higher aspect ratio “a”, which invariably increased the modulus and strength of SFCs. This trend was noticed majorly at lower volume fractions with respect to higher particle size (BB, CC, and DD) variation of HGM.The TGA of SFCs at 5% weight loss shows a 12% reduction in degradation temperature. While at 20% and 55% weight loss, SFCs increased in degradation temperature by 4.5% and 2.7%, respectively. This implies that increased wall thickness “ω” of HGM influenced the TGA of SFCs at higher weight loss;The storage modulus, E’ increased at 30 °C, 50 °C, and 60 °C by 3.2%, 47%, and 96%, respectively. The E’ and E” values of SFCs increased with increasing wall thickness “ω” as the particle sizes increased. The T_g_ determined from tan δ peak at a temperature of above 70 °C increased compared to the NE up to 86 °C;The overall conclusion shows that the flexural strength and bending modulus of syntactic foam were affected by the change in the particle sizes and volume fraction of the HGM filler due to their different interfacial surfaces, wall thicknesses, and aspect ratio. 

## Figures and Tables

**Figure 1 polymers-14-04848-f001:**
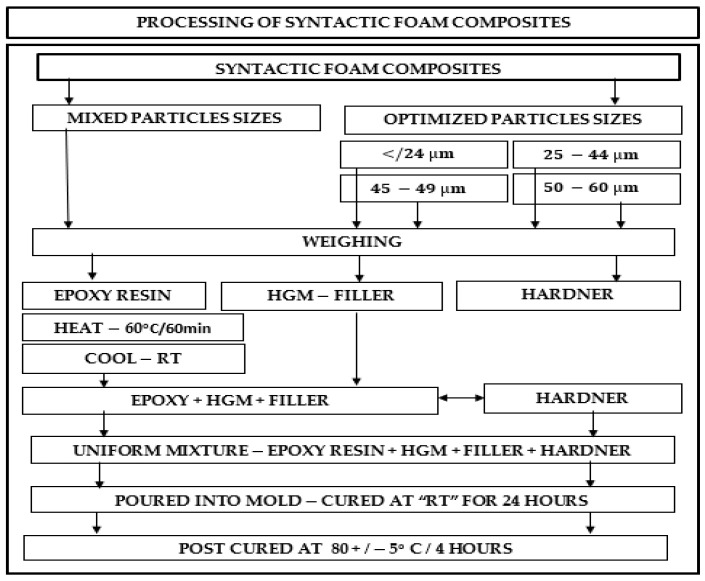
Schematics representation of the syntactic foam formation by the combination of hollow glass microspheres with the epoxy matrix.

**Figure 2 polymers-14-04848-f002:**
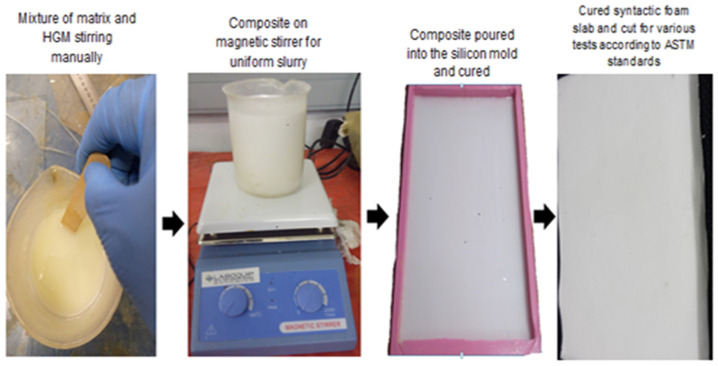
Processing sequence of syntactic foam composite showing different stages involved.

**Figure 3 polymers-14-04848-f003:**
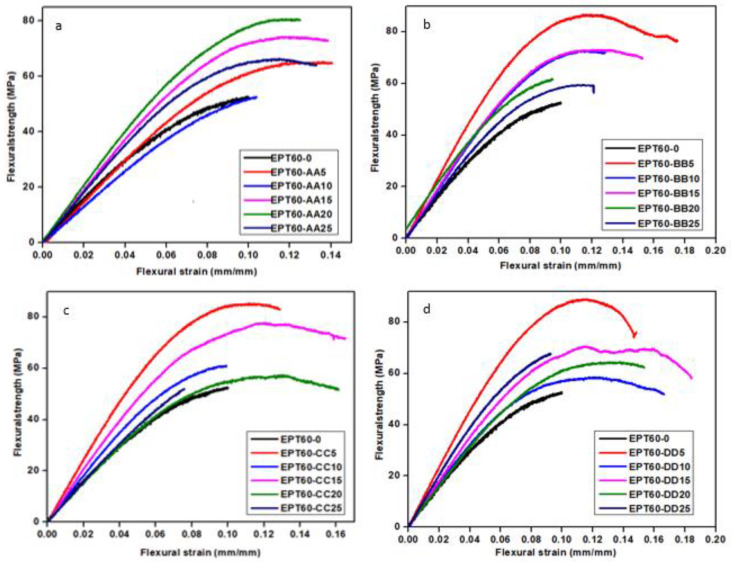
(**a**–**d**): Stress-strain graphs for SFCs compared to the neat epoxy (EPT60-0).

**Figure 4 polymers-14-04848-f004:**
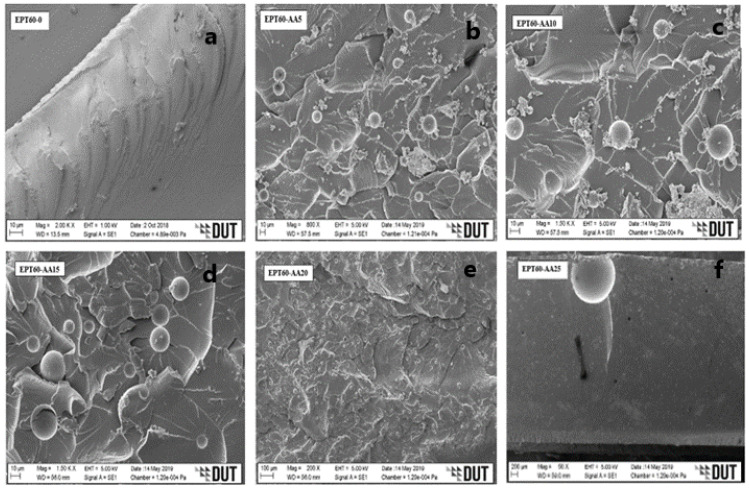
SEM images for flexural fractured specimens of SFC-homogenous HGM at (**a**) EPT60-0, (**b**) EPT60-AA5, (**c**) EPT60-AA10, (**d**) EPT60-AA15, (**e**) EPT60-AA20, and (**f**) EPT60-AA25.

**Figure 5 polymers-14-04848-f005:**
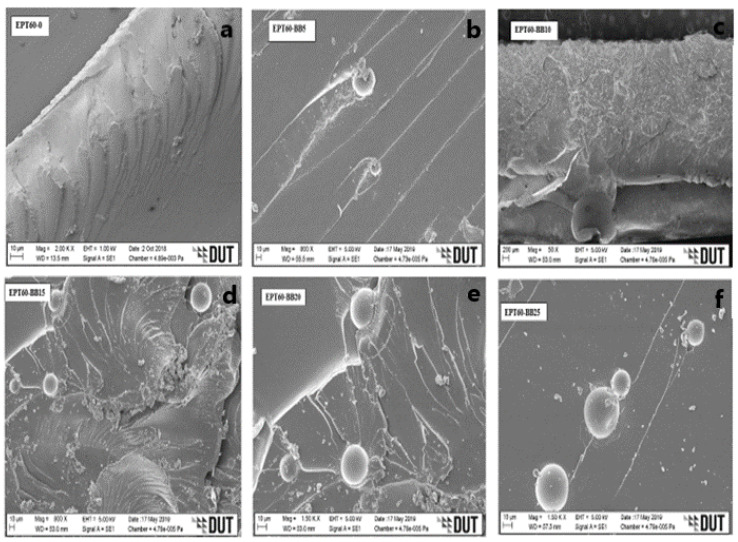
SEM images for flexural fractured specimens of SFC-homogenous HGM at (**a**) EPT60-0, (**b**) EPT60-BB5, (**c**) EPT60-BB10, (**d**) EPT60-BB15, (**e**) EPT60-BB20, and (**f**) EPT60-BB25.

**Figure 6 polymers-14-04848-f006:**
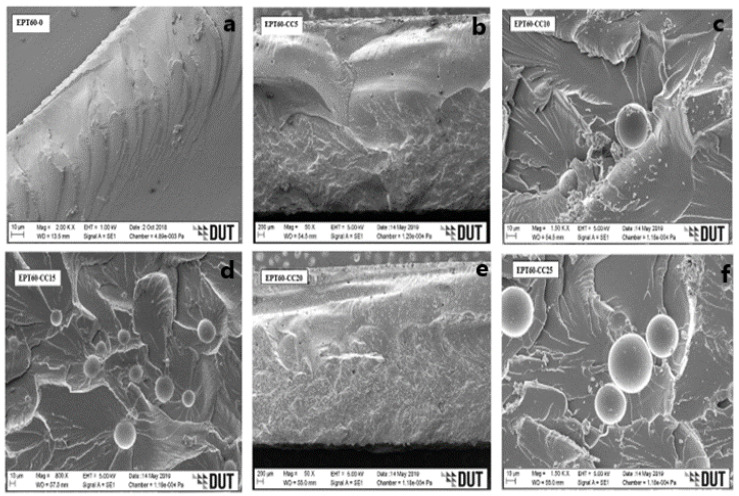
SEM images for flexural fractured specimens of SFC-homogenous HGM at (**a**) EPT60-0, (**b**) EPT60-CC5, (**c**) EPT60-CC10, (**d**) EPT60-CC15, (**e**) EPT60-CC20, and (**f**) EPT60-CC25.

**Figure 7 polymers-14-04848-f007:**
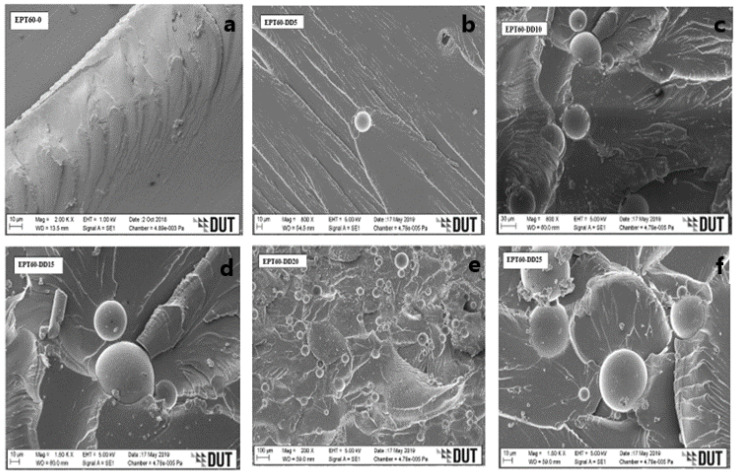
SEM images for flexural fractured specimens of SFC-homogenous HGM at (**a**) EPT60-0, (**b**) EPT60-DD5, (**c**) EPT60-DD10, (**d**) EPT60-DD15, (**e**) EPT60-DD20, and (**f**) EPT60-DD25.

**Figure 8 polymers-14-04848-f008:**
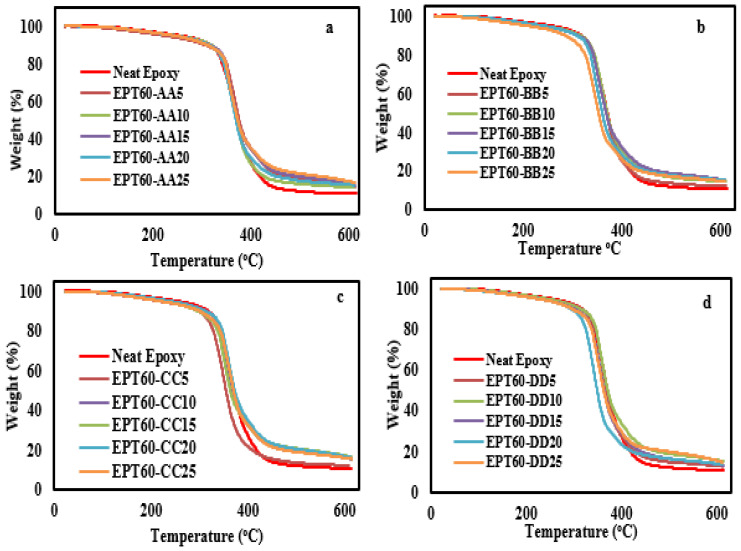
TGA analysis of SFC-homogenous HGM at (**a**) EPT60-AA, (**b**) EPT60-BB, (**c**) EPT60-CC, and (**d**) EPT60-DD. Showing all the volume fraction variations compared with the neat epoxy.

**Figure 9 polymers-14-04848-f009:**
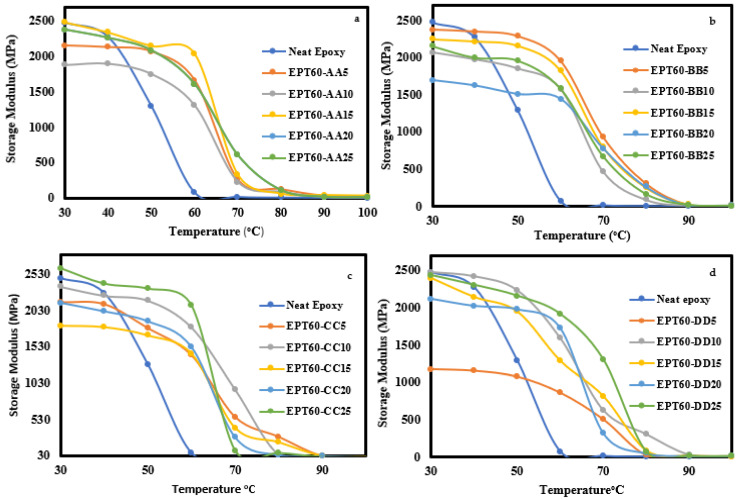
Storage modulus of SFC-homogenous HGM at (**a**) EPT60-AA, (**b**) EPT60-BB, (**c**) EPT60-CC, and (**d**) EPT60-DD. Showing all the volume fraction variations compared with the neat epoxy.

**Figure 10 polymers-14-04848-f010:**
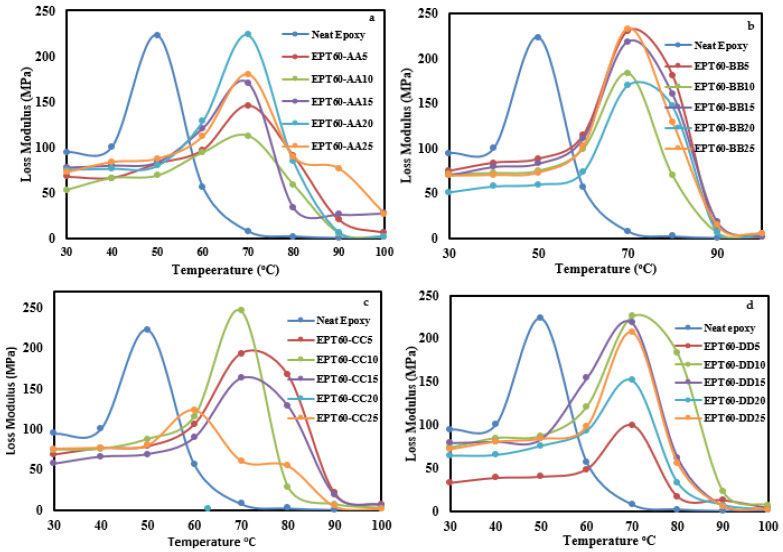
Loss modulus of SFC-homogenous HGM at (**a**) EPT60-AA, (**b**) EPT60-BB, (**c**) EPT60-CC, and (**d**) EPT60-DD Showing all the volume fraction variations compared with the neat epoxy.

**Figure 11 polymers-14-04848-f011:**
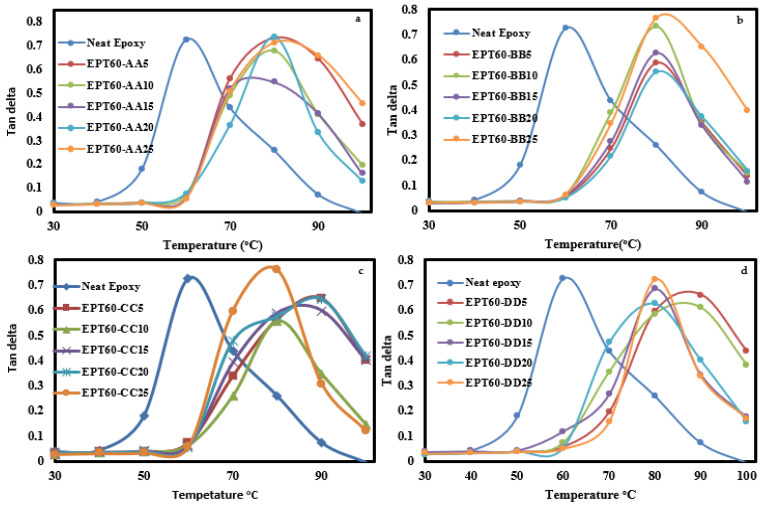
The damping parameter, Tan δ of SFC-homogenous HGM at (**a**) EPT60-AA, (**b**) EPT60-BB, (**c**) EPT60-CC, and (**d**) EPT60-DD. Showing all the volume fraction variations compared with the neat epoxy.

**Table 1 polymers-14-04848-t001:** Particle size distribution of HGM to measure the particle diameter of the size ranges.

HGM Size Variation	D10 Volume (µm)	D50 Volume (µm)	D90 Volume (µm)
AA	5.54	20.59	37.53
BB	5.72	29.39	45.30
CC	6.43	30.42	47.27
DD	14.18	56.07	77.97

**Table 2 polymers-14-04848-t002:** The parameters of particle size distribution of HGM-T60 samples.

Particle Size (µm)	AA	BB	CC	DD
V_d_ (%)	9.66	31.02	9.80	12.16
d (µm)	29.62	30.86	29.40	55.04
ω (µm)	3.22	3.89	4.06	7.67
a	9.58	7.61	7.24	8.23

**Table 3 polymers-14-04848-t003:** Flexural modulus of SFC filled with HGM-epoxy resin (GPa).

Size Variation	Volume Fraction of HGM (%)					
Particle Sizes	NE	EPT60-5	EPT60-10	EPT60-15	EPT60-20	EPT60-25
AA (20–24 µm)	2.4	0.81 ± 0.19	0.60 ± 0.11	1.00 ± 0.13	1.02 ± 0.17	0.94 ± 0.16
BB (25–44 µm)	2.4	1.09 ± 0.19	1.01 ± 0.16	1.02 ± 0.07	0.95 ± 0.13	0.82 ± 0.05
CC (45–49 µm)	2.4	1.31 ± 0.25	0.81 ± 0.14	1.06 ± 0.16	0.74 ± 0.20	0.90 ± 0.13
DD (50–60 µm)	2.4	1.14 ± 0.20	1.01 ± 0.11	0.86 ± 0.10	0.78 ± 0.14	1.03 ± 0.08

**Table 4 polymers-14-04848-t004:** Flexural strength of SFC filled with HGM-epoxy resin (MPa).

Size Variation		Volume Fraction (%)				
Particle Sizes	NE	EPT60-5	EPT60-10	EPT60-15	EPT60-20	EPT60-25
AA (20–24 µm)	53.5	65.3 ± 8.3	52.6 ± 0.6	74.4 ± 4.8	80.7 ± 9.2	66.3 ± 9.1
BB (25–44 µm)	53.5	86.7 ± 3.5	72.7 ± 3.6	70.7 ± 2.2	62.8 ± 6.5	59.5 ± 4.2
CC (45–49 µm)	53.5	85.5 ± 2.6	61.2 ± 5.4	77.8 ± 7.2	57.6 ± 2.9	52.1 ± 0.9
DD (50–60 µm)	53.5	89 ± 5.1	58.6 ± 3.6	70.5 ± 2.0	64.5 ± 7.8	67.8 ± 1.1

**Table 5 polymers-14-04848-t005:** Flexural strain of SFC filled with HGM-epoxy resin (mm/mm).

Size Variation	Volume Fraction (%)					
Particle Sizes	NE	EPT60-5	EPT60-10	EPT60-15	EPT60-20	EPT60-25
AA (20–24 µm)	0.044	0.141 ± 0.014	0.104 ± 0.006	0.139 ± 0.014	0.125 ± 0.018	0.133 ± 0.021
BB (25–44 µm)	0.044	0.175 ± 0.011	0.128 ± 0.005	0.153 ± 0.022	0.114 ± 0.042	0.212 ± 0.055
CC (45–49 µm)	0.044	0.129 ± 0.023	0.099 ± 0.020	0.165 ± 0.056	0.193 ± 0.059	0.076 ± 0.034
DD (50–60 µm)	0.044	0.148 ± 0.034	0.167 ± 0.029	0.225 ± 0.049	0.154 ± 0.011	0.093 ± 0.102

## Data Availability

Not applicable.

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
