# Peer review of "Analysis of Particle Variation Effect on Flexural Properties of Hollow Glass Microsphere Filled Epoxy Matrix Syntactic Foam Composites"

_polymers, 2022, doi:10.3390/polym14224848_

Round 1

Reviewer 1 Report (Previous Reviewer 1)

This re-submitted manuscript reported by Afolabi et al has showed more experiments and detailed data analyses, thus displaying greatly improvement in describing their research results. Indeed, according to the experimental results, it is obvious that changing the particle sizes and volume fraction of the HGM filler would cause different interfacial surfaces, wall thicknesses, and aspect ratio, affecting the flexural strength and bending modulus of syntactic foam. The manuscript is now acceptable for publishing in Polymers. Specific comment(s):   

1.     Resolution of Figures 8-11 should be improved.

2.     In abstract, “…30C, 50C, and 60C…” should be “…30°C, 50°C, and 60°C…”.

Author Response

Reviewer 2 Report (New Reviewer)

Paper can be considered for publication after several corrections:

1)      Introduction, last paragraph: “This study focused on the effect of different particle sizes, aspect ratio and wall thickness on the flexural strength and modulus properties of syntactic foam composite.” Indeed, no effect of either aspect ratio or wall thickness on the composite properties was considered in paper, thus this sentence should be corrected.

2)      Section 2.1. Raw HGM should be characterized in detail here.

3)      Section 2. How the HGM fraction was separated? Please describe the procedure in detail.

4)      Tables 3 and 5. Please give the standard deviations.

Round 2

Reviewer 2 Report (New Reviewer)

Paper can be accepted now

This manuscript is a resubmission of an earlier submission. The following is a list of the peer review reports and author responses from that submission.

Round 1

Reviewer 1 Report

This manuscript reported by Afolabi et al has mainly studied the effect of particle size variation on the mechanical properties of hollow glass microsphere/epoxy resin (HGM/EP) syntactic foam composites (SFC). Meanwhile, the influence of volume fraction of HGM on the flexural strain at the break of the SFC has also been studied. The microstructural analyses on the fracture surfaces of HGM/EP SFC have also been done by SEM images. The study clearly concluded that the flexural stress and strain of the syntactic foam composites increased at the inclusion of the HGM filler more than the neat epoxy for all the size variations and volume fractions except at EPT60-AA10 and EPT60-CC25. The main claim to improve the flexural strength compared to neat epoxy is due to a better dispersion of filler particles in the matrix during mixture and the good surface interaction occurred between the micrograph of the HGM and the epoxy resin with little agglomeration. However, the manuscript is not recommended to be publishing in Polymers because of the following concerns:

1.      All the abbreviations in the abstract must be defined.

2.      All the HGM/EP SFCs possess good bending capacity than the EPT60-0 except EPT60-AA10. The authors claimed that this is due to the poor interaction between the filler and the matrix during mixing. However, they did not tell us why only the case possessed poor filler-matrix interaction while others did not?

3.      The stress-strain relationships show only a clear trend in EPT60xx-BByy series, but are very random in the EPT60xx-AAyy, EPT60xx-CCyy, and EPT60xx-DDyy series, such results do not tell us useful information how the particle size variation and volume fraction of HGM affect the flexural modulus of SFC. The data should be checked carefully or the phenomena should be explained clearly.

4.      The authors use both EPT60-0 and NE to represent neat epoxy, please use only one to avoid confuse.  

Reviewer 2 Report

Comments to authors are listed below:

1.      Abstract should be enriched via valuable results which pave the way for understanding the audiences.

2.      The introduction is poor and written with poor information and did not cover the importance of this topic. So, it should be extended with further recent literature.

3.      The authors are requested to include a detailed and comprehensive study of an application and report the novelty of this work in the introduction section.

The investigations by using two characterisations (Tensile and SEM tests) are limited and further relevant tests should be performed to reach the high quality papers, for example, TGA and DSC, and DMA  tests.

4.      Conclusion is very short and lack the basic fundamentals of the results obtained. Please, authors should re-write the conclusions again  with more emphasis on the significant comparison and the improvements from the results obtained.  

5.      The English writing for whole paper are required to be improved.